# The 2024 Noto Peninsula earthquake building damage dataset: Multi-source visual assessment

Ruben Vescovo[1], Bruno Adriano[2], Sesa Wiguna[1], Chia Yee Ho[1], Jorge Morales[1], Xuanyan Dong[1], Shin Ishii[1], Kazuki Wako[1], Yudai Ezaki[3], Ayumu Mizutani[2], Erick Mas[2], Satoshi Tanaka[4], and Shunichi Koshimura[2]

[1]Department of Civil and Environmental Engineering, Tohoku University, Aoba 468–1, Aramaki, Aoba-ku, Sendai, 980–8572, Japan

[2]International Research Institute of Disaster Science (IRIDeS), Tohoku University, Aoba 468–1, Aramaki, Aoba-ku, Sendai, 980–8572, Japan

[3]Department of Civil Engineering and Architecture, School of Engineering, Aoba–6–6–06 Aramaki, Aoba Ward, Sendai, Miyagi 980–8572

[4]Faculty of Social and Environmental Studies, Department of Social and Environmental Studies, Tokoha University, Yayoi-cho 6–1, Suruga-ku, Shizuoka city, 422–8581, Japan

**Correspondence:** Shunichi Koshimura (shunichi.koshimura.a3@tohoku.ac.jp)

**Abstract.** We present a building damage dataset following the 2024 Noto Peninsula Earthquake. The database was compiled from freely available, multi-source, remote sensing data, verified through opt-in crowd-sourced information. The dataset consists of geo-referenced polygons representing the pre-event building footprints of 140,208 structures. Each building was classified through visual inspection using pre-disaster and post-disaster vertical, oblique, survey, and verifiable news reporting imagery. Entries were validated using voluntary-submission data sourced through a web-API hosting a live version of the database. We calculate classification metrics for a subset of the database where ground survey photographs were provided by independent surveyors. An average $F_1$-score of 0.94 suggests that the proposed assessment is consistent and high quality. We aim to inform future research such as disaster-specific physical dynamics models; statistical and machine learning damage models; logistics and evacuation studies. The present work describes the data collection process, damage assessment methodology, and rationale; including limitations encountered, the crowd sourcing validation process, and the dataset structure.

## 1 Introduction

At 16:10 January 1st 2024, shallow reverse faulting produced a $M_w7.5$ earthquake (USGS, 2024) that propagated from the north-most point of Suzu City, Ishikawa Prefecture (Figure: 1). The disaster poses unique challenges for the disaster geophysics community due to its context and outfall. The intraplate faulting occurred on the relatively inert western coast of Honshu Island, Japan, following a 3 year long earthquake swarm (Ishikawa and Bai, 2024). Affected areas were the Suzu, Noto, Wajima, Nanao, Shika and Anamizu municipalities (NASA, 2024; Japan Meteorological Agency, 2024; Japanese Red Cross Society, 2024). The Prime Minister's Office of Japan has provided transcripts for several press conferences and emergency meetings reporting actions taken to address monitoring and relief operations. Initial reporting informed of near instant tsunami

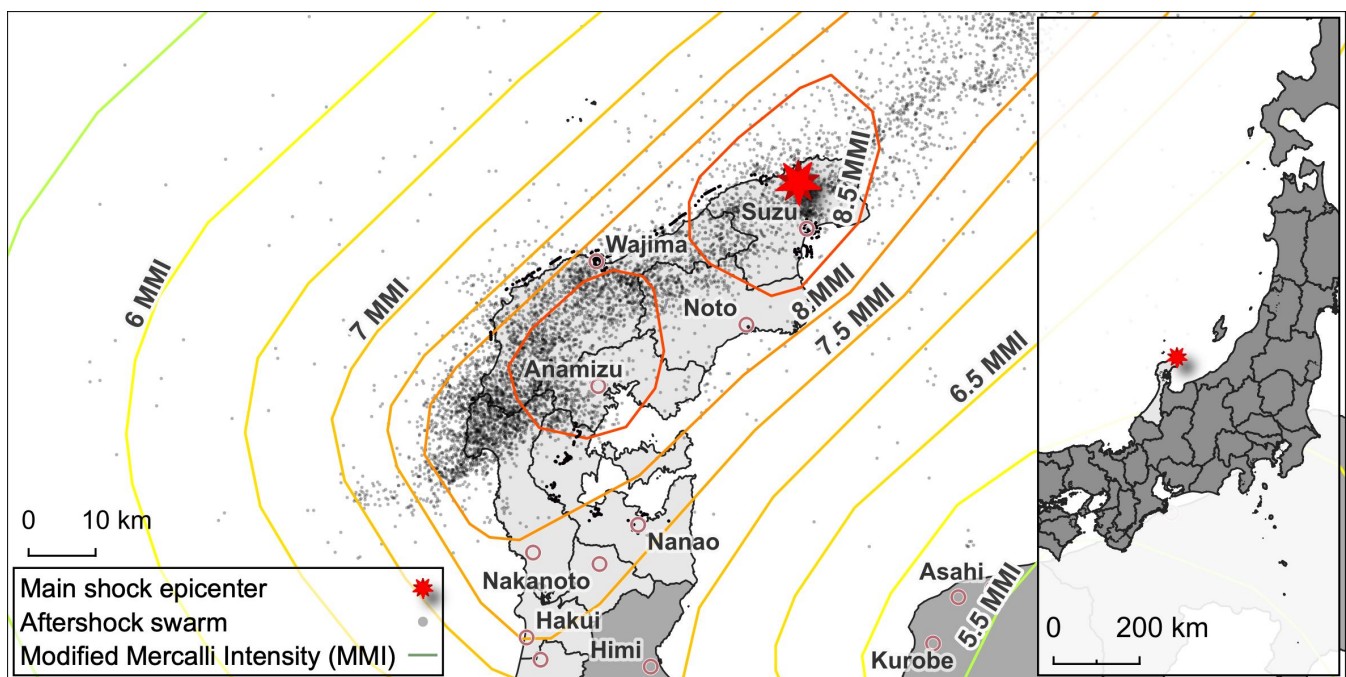

**Figure 1.** Seismic context of the 2024 Noto Peninsula Earthquake showing the distribution of the earthquake swarm following the aftershocks (USGS, 2024; GSI, 2024).

impacts around the main shock's epicenter (north Suzu), quickly followed by a comprehensive tsunami warning along the entire
peninsula's coast. Subsequent statements confirmed the presence of catastrophic damage affecting infrastructure throughout
the peninsula including, ground shaking, land deformation, liquefaction, and landslides causing varied damage to buildings,
interrupting roads, originating a fire. Later press releases, medical reports, and news outlets confirmed impacts to critical ser-
vices, such as water supply, sewage system, power outages, and telecommunication service disruptions (Egawa et al., 2024;
Prime Minister's Office of Japan, 2024; Prime Minister's Office of Japan, 2024a, b, c; British Broadcasting Corporation, 2024).
The disaster and following outfall ultimately resulted in injuries and human casualties, the prevention of which represents an
overarching focus of disaster research (preventable disaster deaths) (Egawa et al., 2024).

The spatial distribution of infrastructure impacts is of particular importance to disaster research. Such data can inform emer-
gency response studies (physical dynamics simulation, damage detection, damage estimation, evacuation simulation, etc.), long
term recovery studies (socioeconomic studies, disaster epidemiology, disaster prevention, probabilistic hazard, etc.), and ulti-
mately the development of more informed codes and regulation. Disaster damage visual assessments are critical to develop a
comprehensive corpus of disaster impacts to infrastructure and to inform studies such as the aforementioned. These visual as-
sessments can be carried out by an on-the-ground survey team ensuring the highest degree of fidelity and granularity. However
such an investigation is often resource intensive, carries inherent risk of harm, and may be highly invasive. Alternative meth-
ods generally employ remote sensing data and have historically been carried out by experts (Gokon and Koshimura, 2012) and

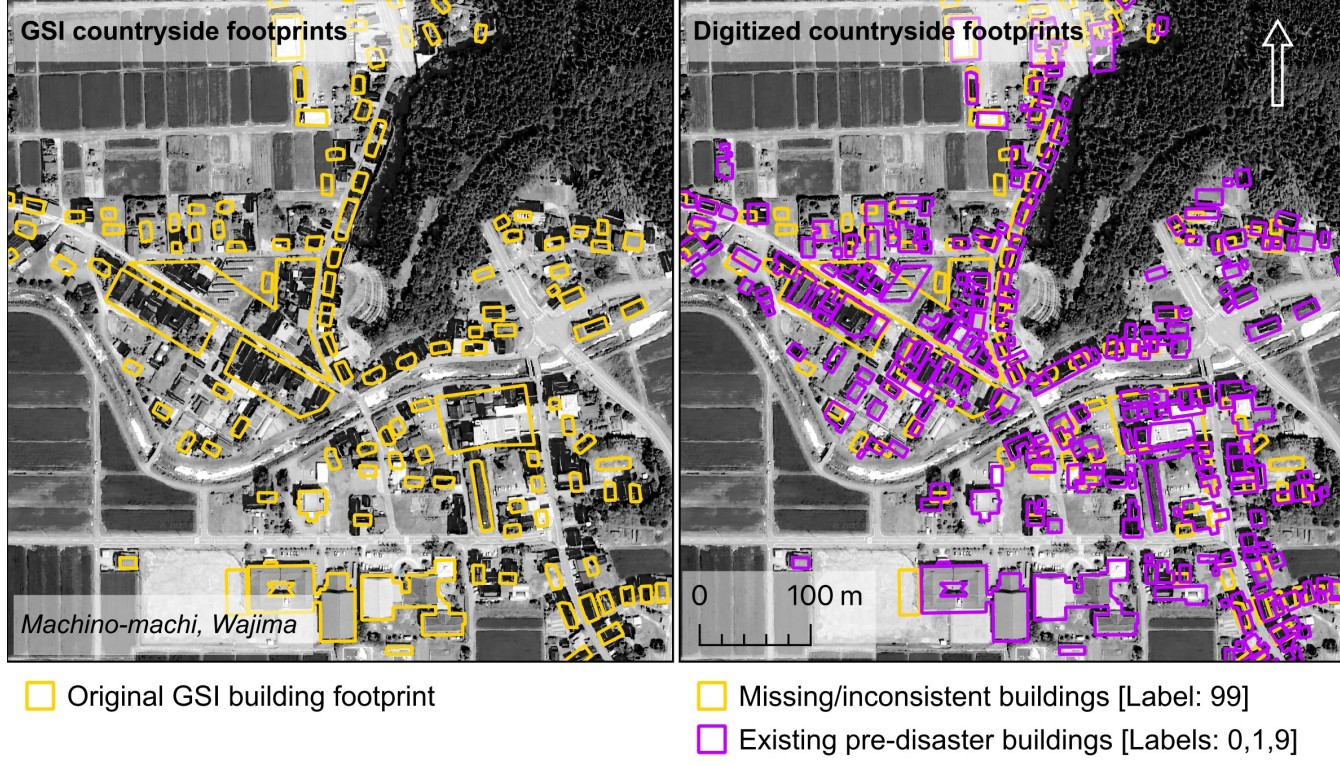

| | |
|---|---|
| ☐ Original GSI building footprint | ☐ Missing/inconsistent buildings [Label: 99] |
| | ☐ Existing pre-disaster buildings [Labels: 0,1,9] |

**Figure 2.** Mismatch between original GSI-sourced building polygons and orthophoto imagery. Inconsistencies between the GSI-sourced building polygons and the orthophoto imagery were observed in rural areas. As part of the classification process, newly identified buildings were added to the database, while existing building footprints were adjusted to match the orthophoto imagery. In case of ambiguity, an existing GSI building is marked as 99 (missing or inconsistent) and replacing buildings are redrawn before being classifies as described in Section 2 (Basemap attribution: © Google, 2024 Airbus, CNES/Airbus, Japan Hydrographic Association, Landsat/Copernicus, Maxar Technologies 2024).

institutions (CEMS, 2025). Human visual assessments have informed several studies that have contributed to a deeper understanding of seismic and tsunami building damage. Chua et al. (2021) for example conducted a limited scope visual assessment of the 2011 Great East Earthquake and Tsunami using multi-source multi-modal imagery (Modality refers to the viewing angle, including aerial or satellite orthophoto, standard aerial photography, ground level photo, or aerial oblique photos, among others. Source refers to the type and capabilities of the sensor used to capture the imagery) to generate fragility curves of port structures in Ishinomaki City. More recently automated methods employing pre-trained machine learning models have been explored (Deng and Wang, 2022; Miura et al., 2020; Wiguna et al., 2024a). Such methods generally leverage vertical imagery as input to a machine learning model to perform automatic classification of building damage. These automated image based assessments carry inherent limitations beyond human interpretation, such as the capability to generalize between domains and

hazard induced damage (e.g. damage to buildings from tsunami, earthquake, fire, etc. all look different). A preliminary in-
45 vestigation of the building footprint inventory revealed large discrepancies with pre-event imagery (Figure 2). Moreover, the
variable aerial survey periods, image capture quality, meteorological conditions (Table 1), and different mechanisms driving
building failure (Such as fire, tsunami, earthquake, etc.; Figure 3) contributed to a visually fragmented and inconsistent domain.

| Date taken | GSI mosaic name | Notes |
| --- | --- | --- |
| 2024–01–02 | Suzu | modest overcast (east), inland snow buildup (mild) |
| | Wajima-Naka | mostly overcast, inland snow buildup (modest) |
| | Wajima-Higashi | minimal overcast (east), inland snow buildup (mild) |
| 2024–01–05 | Suzu | mild overcast (west), otherwise clear |
| | Nanao | minimal overcast (southwest), generally clear, heavy desaturation |
| | Anamizu | major overcast, central coast clear |
| 2024–01–11 | Wajima-Naka | minimal overcast (southeast), snow buildup (modest) |
| | Anamizu | minimal overcast (center), inland snow buildup (modest) |
| | Wajima-Nishi | mild overcast (south), snow buildup (modest) |
| 2024–01–17 | Nanao | clear, snow buildup (heavy), slight desaturation |
| | Wajima-Nishi | clear, inland snow buildup (modest), coast snow buildup (mild) |
| | Anamizu | clear, snow buildup (heavy) |

**Table 1.** Characteristics and date of each GSI vertical mosaic

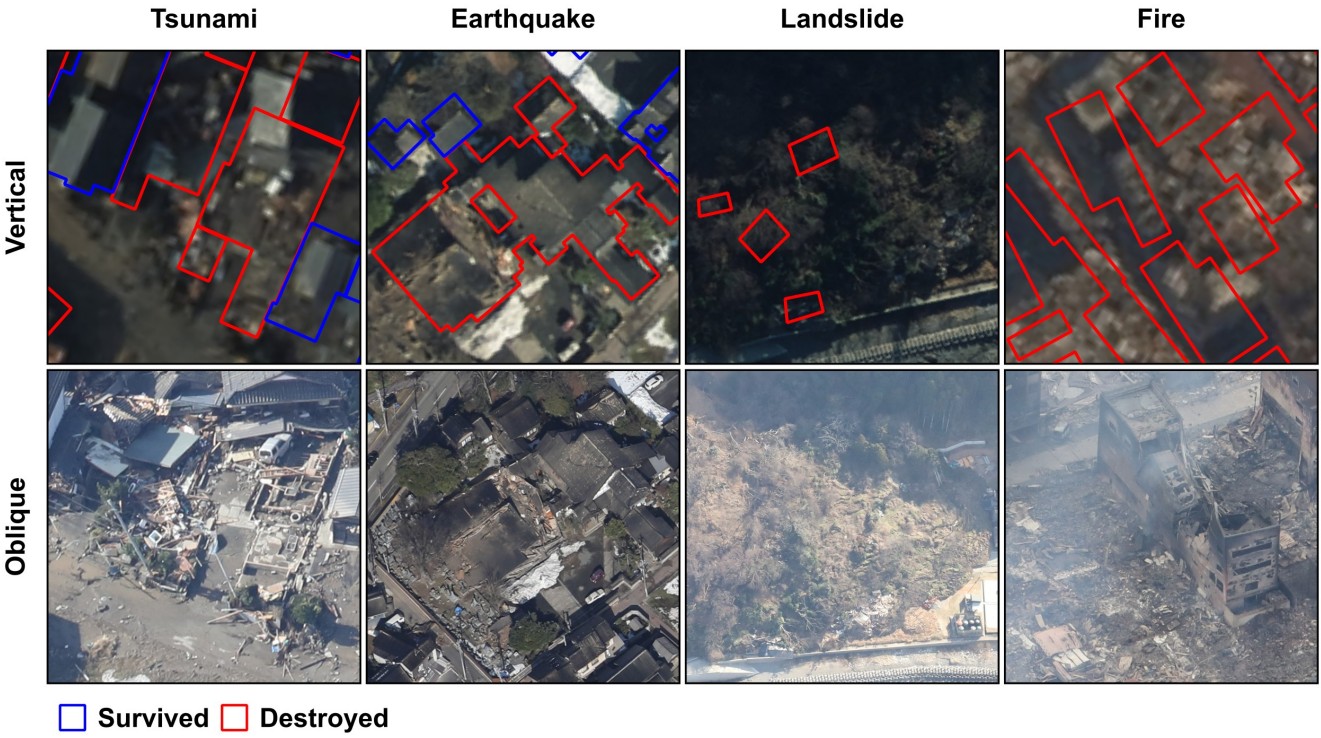

**Figure 3.** Building damage across the Noto Peninsula is split between failure modes consequent to different, often compounding hazards. Vertical aerial imagery by GSI (2024), Oblique imagery by © KKC (2024)

Despite these challenges, the unprecedented availability of open-source & multi-source data provided us with a unique
opportunity for a rapid visual damage assessment. With the above considerations in mind, we opted for a manual approach
to curate this dataset. We hope that this dataset will serve as a reference for future studies and as a benchmark for automated
methods. The primary sources for the investigation were post-disaster vertical imagery captured by the Geospatial Information
Authority of Japan (GSI) and made available online. In addition oblique imagery of select portions of Noto Peninsula were made
available by Kokusai Kogyo (KKC) through the free version of their proprietary aerial survey database (for details regarding
licensing, usage, and distribution, see Section 6). The post-disaster imagery data informed the classification of the public GSI
building footprint inventory vector data. Our criteria was developed iteratively in response to limitations presented by the
data. Following an initially limited-scope investigation, the assessment was made available to the public for a progressive
appraisal at the online portal: https://experience.arcgis.com/experience/70aae9964dc54e4190b6b360dcbb3759/. End users
may request corrections regarding potential misclassifications. Requests must include proof to substantiate the amendment,
usually in the form of a photo of the target building. Finally, two limited-scope on-site surveys by independent research teams
informed a secondary round of corrections. We hope to contribute to the growing corpus natural hazard driven building damage
datasets, in the manner of the 2009 L'Aquila Earthquake Dataset (Tertulliani et al., 2012) and the 2011 Great East Japan

Earthquake Dataset (Sekimoto et al., 2013) which have been widely studied and applied downstream to advance the field of disaster research (Anniballe et al., 2018; Suppasri et al., 2014; Charvet et al., 2014). The initiative takes inspiration from previous efforts to democratize this process such as Tomnod (DigitalGlobe and formerly GEO-CAN) for cases like Haiti (Zhai et al., 2012), Christchurch (Barrington et al., 2012; Ghosh et al., 2011), and Nepal (Poiani et al., 2016), who in different ways employed crowd source techniques to supplement the damage assessment process. Beyond just contributing more data, we hope to inform future research in three fundamental ways:

- Prove the feasibility of multi-modal, multi-source visual assessment methodologies;

- Provide a measure of expected accuracy when employing such methods; and

- Contribute a damage dataset of a profoundly complex disaster, with significant multi-hazard interactions and impacts.

## 2 Methods

Herein, we relate the methodology used to generate the dataset, including: considerations, challenges, and limitations encountered during the creation of the dataset (Figure 4). The working group was formed in response to the disaster and included members from Tohoku University, the International Research Institute of Disaster Science (IRIDeS), and the Faculty of Social and Environmental Studies at Tokoha University who provided a secondary survey sample to conduct the technical validation. The assessment was conducted by our internal working group, which included a mix of civil engineers, geophysicists, and disaster researchers. We conducted the assessment in a collaborative manner, with each member of the working group contributing to the assessment of different areas of the peninsula, followed by a quality control round to ensure consistency. The assessment progress was publicly documented through a web portal, which allowed for real-time feedback from the public and other researchers, as more parts of the peninsula were documented and served online. After the initial assessment, we conducted a secondary round of validation using limited-scope surveys by independent research teams, and crowd-sourced feedback from the public.

### 2.1 Data sourcing

The initial review consisted of a general overview of available data from official sources. The Government of Japan provides basic geographic information through the GSI (https://fgd.gsi.go.jp/download/menu.php, Japanese). An inventory of building footprints, pre-event aerial imagery, and digital elevation data, provided a general level of clarity for the feasibility of a visual assessment. Moreover, GSI hosts an index of information pertinent to the Noto Peninsula Earthquake on a dedicated page (GSI, 2024) (https://www.gsi.go.jp/BOUSAI/20240101_noto_earthquake.html, Japanese) at time of writing. Available data includes: ground subsidence and slope failure extents, post-event aerial vertical imagery, tsunami inundation extent estimates, and crustal deformation estimates. From this portal we obtained post-processed vertical imagery xyz tiles for the post disaster period: GSI conducted photographic missions on January 2nd, 5th, 11th, and 17th, covering the whole Noto Peninsula with a significant degree of redundancy to minimize visual obstruction due to atmospheric and environmental effects such a cloud coverage, smoke,

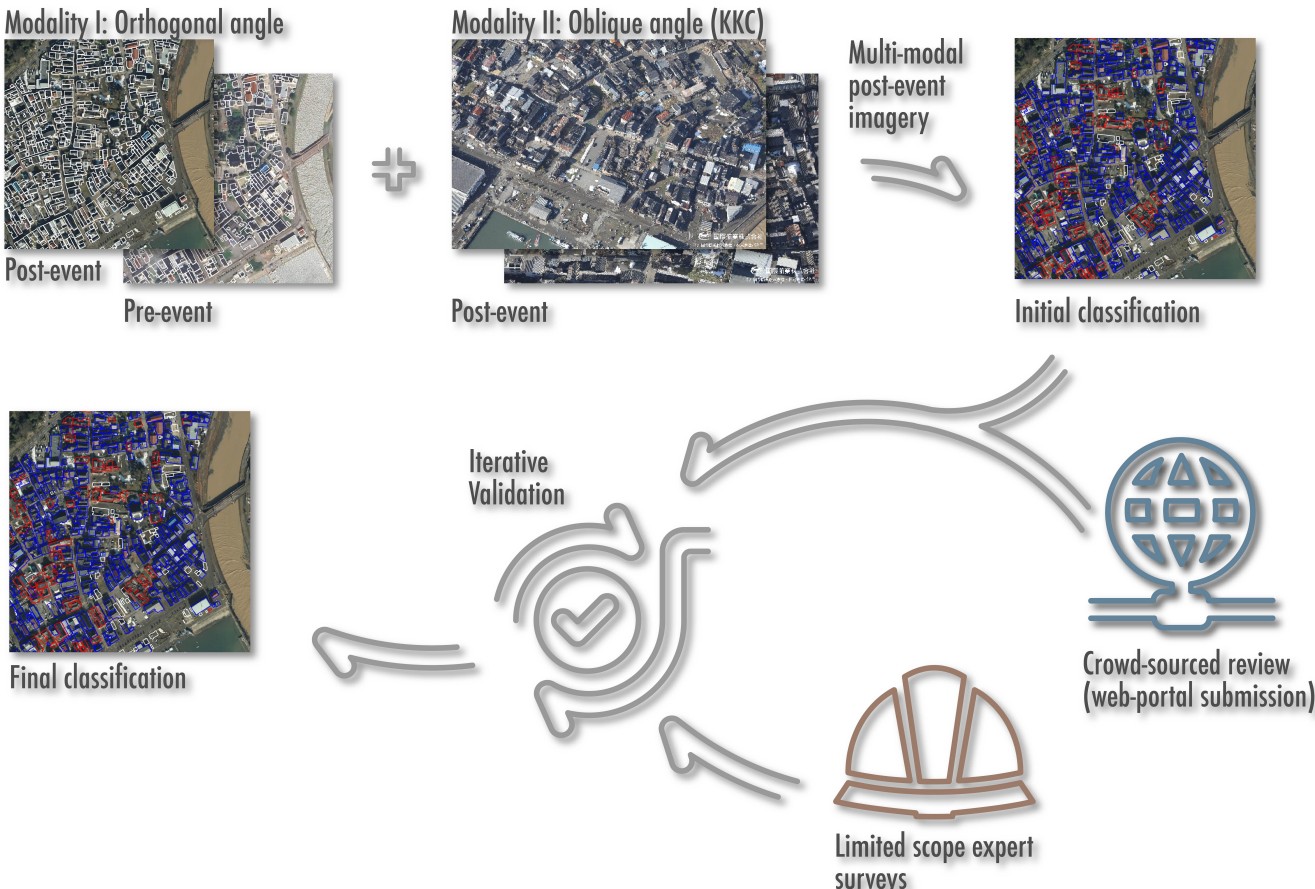

**Figure 4.** Building damage visual assessment workflow illustrating the working group's approach to multi-source & multi-modal data, each stage of inclusion and expert-feedback-driven iterative validation process. Pre-event orthophoto by © Google 2024, Post event aerial by GSI (2024), All oblique imagery courtesy of © KKC (2024).

sunshade, and snow. Similarly, Kokusai Kogyo (KKC), a for-profit consulting agency specializing in geospatial technology, has been issuing special investigative products free of charge through a Noto Peninsula Earthquake information page (KKC, 2024). Available data includes geotagged high-resolution oblique imagery of specific high-profile areas that were affected catastrophically by the disaster (KKC, 2024); oblique imagery missions were conducted on January $2^{nd}$; Oblique images are available through Kokusaki Kogyo's proprietary BOIS portal https://bois-free.bousai.genavis.jp/diarsweb (Japanese). The assessment was supplemented by news sources for select areas of Nanao City, due to visibility issues with the vertical and oblique images (Table 1)(Minami, 2024; xTECH, 2024).

## 2.2 General Methods

We provide a short summary of the general methodology used to conduct the visual assessment (Refer to Table 2 for precise criteria):

1. areas with oblique images were treated first.

2. Each aerial was screened for cloud cover and potentially obstructed buildings were marked as 9 unless the oblique imagery provided a clear view of the building, in which case the oblique was used to classify the building;

3. for each building we checked first against the pre-disaster aerial for footprint geometry consistency, in case of mismatch or ambiguity we classified the existing polygon as 99 and added a new polygon (based on the pre-disaster aerial) to the database;

4. in case of coherence between the footprint polygon and the pre-disaster aerial, we classified the building as 0 or 1 (survived/destroyed) using the post-disaster vertical imagery; and corroborated against the oblique image if available and clear.

5. After each section was completed, we conducted a quality control round to ensure consistency across the assessment.

6. After the initial assessment, we conducted a secondary round of validation using limited-scope surveys by independent research teams, and crowd-sourced feedback from the public.

The assessment began as a limited-scope pilot investigation for the tsunami affected area, but expanded to include the entire peninsula. Based on the pilot investigation, we conceived an initial "binary+" classification schema that was eventually formalized into the final classification system with minimal adjustment - these classes are defined as reported in Table 2. We provide an approximate equivalence table between classification methods in Table 3. This decision is in part supported by previous findings by Huynh et al. (2014) who mention that crowd-sourcing yields bias towards edge classes ("No Damage" and "Destroyed") in spite of middle classes (Damage and/or Possibly Damaged). Where possible, the visual assessment is supported by oblique imagery, which proved invaluable in many instances. This was especially true for edge cases such as areas with poor visibility, densely packed areas wherein buildings collapsed vertically ("pancake" collapse), or overcast areas. Initially we considered a multi-class damage assessment to fully leverage the oblique imagery. However, a combination of cloud obfuscation in the vertical imagery and limited coverage of the obliques conditions disallowed a comprehensive assessment (Figure 5). As mentioned in Section 1, there exist significant discrepancies between the GSI building inventory and the pre-event orthophotos. These discrepancies are particularly pronounced in rural areas where the GSI building inventory is often missing or inconsistent with the orthophoto baseline. These mismatches range from minor misalignments, to geometry changes (in the case of additions or refurbishing), to significant changes in the building footprint (in the case of demolitions or new constructions), we highlight an example in Figure 2. When reasonable, we attempted to adjust the GSI building footprints to match the orthophoto baseline, this is done in order to preserve the original GSI building metadata (see Section 3). However, in most cases the ambiguity is

| Class label | Database value | Criterion |
|---|---|---|
| 0 | Survived | Damage does not appear to affect the bearing mode of the structure - Includes:<br>• Partial damage of the roof requiring replacement or repair.<br>• Buildings in the vicinity of structurally unsound buildings but appear structurally sound.<br>• Undamaged buildings. |
| 1 | Destroyed | Structurally unsound based on visual interpretation - Includes:<br>• Partially or completely washed away buildings.<br>• Partially, completely collapsed, or severely inclined buildings.<br>• Partially or completely buried buildings.<br>• Buildings burned to the degree that they are structurally unsound. |
| 9 | Obstructed view | Building is marked by a footprint according to the GSI registry but is visually obstructed - Includes:<br>• Buildings under cloud cover.<br>• Buildings under sunshade such that they are indistinguishable from their surroundings.<br>• Buildings under canopy cover such that structural features are indistinguishable. |
| 99 | Missing or inconsistent | Buildings that whose GSI registry footprint is significantly inconsistent relative to available imagery - Includes:<br>• Building footprints that do not match an existing building across pre-event and post-event imagery even when allowing for a degree of vertical shift.<br>• Building footprints that demarcate a non-existing building across pre-event and post-event imagery. |

**Table 2.** Criteria used for binary classification of the entire Noto Peninsula building damage visual assessment.

significant enough that the building is marked as 99 (missing or inconsistent, Table 2) and a new buildings are drawn based in its place.

| Damage Grade (Grünthal et al., 1998) | Damage Index (Okada and Takai, 1999) | Damage Index acronyms | Copernicus EMS (CEMS, 2025) | Present Study |
|---|---|---|---|---|
| D0 | 0.0 | Nd0 | Possibly damaged | Survived |
| D1 | 0.1~0.2 | Md1 | Damaged | |
| D2 | 0.2~0.4 | Md2 | | |
| D3 | 0.4~0.6 | Ud3, Gd3, Ed3, Rd3, Sd3 | | |
| D4 | 0.6~0.8 | Ud4, Gd4, Ed4, Sd4 | Destroyed | Destroyed |
| D5 | 0.8~0.9 | Ud5-, Ud5+, Gd5-, Gd5+, Sd5 | | |
| | 0.9~1.0 | Cd5+ | | |

**Table 3.** Comparison of popular reference damage scales for building damage visual assessment with approximate relative equivalences.

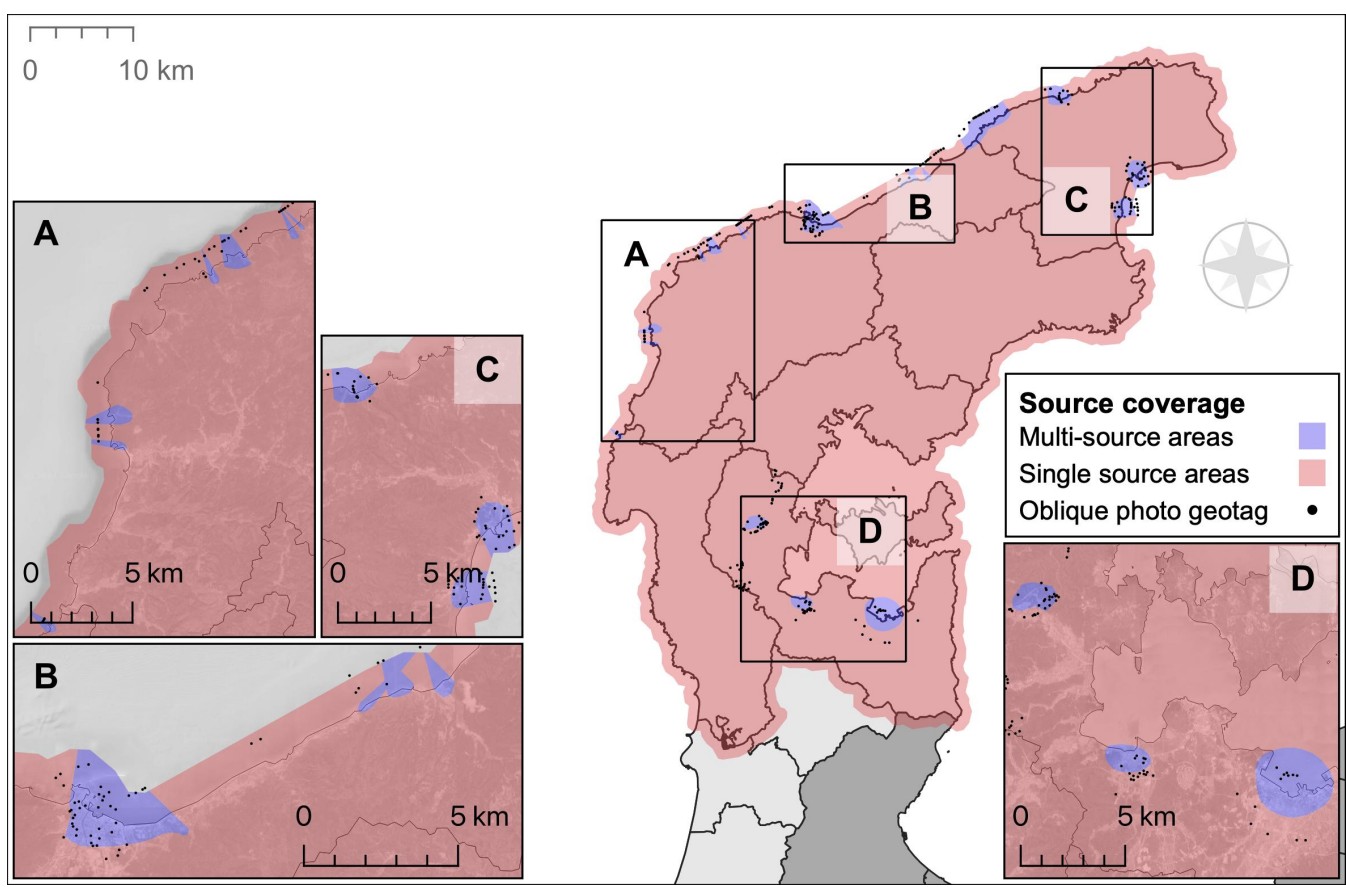

**Figure 5.** Oblique coverage by Kokusai Kogyo (KKC, 2024) and inherent confidence of visual assessment. Inset basemap ©  Google 2024

Since only $\sim 16\%$ of buildings lie within the viewing angle of oblique imagery a multi-class assessment was deemed less viable. Miura et al. (2020) make a case for the inclusion of blue-tarp covered buildings as a separate class in their deep learning classification framework: they notice that presence of blue-tarp covered structures correlated with moderate-heavy building damage classes. Although we initially considered including tarp-covered buildings as a distinct class, there exist mismatches between the GSI provided vertical images: this is observable for segments where overlapping orthophotos, such as Wajima City, are available (A schedule of vertical imagery acquisition missions conducted by GSI is provided in Table 1). Figure 6 provides an example of the mismatch in tarp presence between mission dates. In other instances, such as Anamizu-machi, spotty cloud cover makes the identification of tarp-covered buildings particularly challenging. Ultimately, a conservative approach was deemed preferable.

## 2.3 Tsunami damage assessment

The tsunami that impacted the eastern coast of the Noto Peninsula was purportedly generated in part by the rupturing of several offshore active faults. In addition, seismic activity may have aggravated submarine landslides in southern Toyama Bay, leading to subsequent tsunami amplification that was ultimately responsible for much of the damage experienced in along the eastern coast of the Noto Peninsula (Masuda et al., 2024). The estimated tsunami inundation extent lies almost entirely along the northeastern coast of the Noto Peninsula area and stretches from the northern most point of the Suzu municipality, to the Nanao municipality in the south. Yuhi et al. (2024a, b) conducted several surveys of the tsunami inundation area and provided comprehensive information on the inundation and run-up heights of the tsunami. On the western coast, only a small extent on the northern portion of the Shika municipality was indicated as inundated (Figure 7). The intersection between the estimated tsunami inundation and the GSI building inventory was the first portion of the damage assessment to be carried out as a preliminary measure. 3,261 were originally included, however significant mismatch exists between the GSI footprints and the orthophoto base map - particularly in non urban areas. Mismatches have been handled as noted in Table 4.

| Case | | Action |
|---|---|---|
| Polygon does not reflect the shape in the orthophoto | $\rightarrow$ | Adjust (add, split, merge) |
| Polygon does not appear to correspond to a pre-event or post-event building | $\rightarrow$ | Mark building as 99 |
| Polygon does not exist, building is evident on pre-disaster orthophoto | $\rightarrow$ | Polygon is manually added |

**Table 4.** Approach to mismatches between footprint polygons and the orthophoto baseline.

## 2.4 Earthquake damage assessment

The scope of the visual assessment was expanded upon completion of the tsunami assessment. The criteria was adjusted to include modes of damage exogenous to tsunami induced failure: including considerations for landslide displacement & burial, fire damage. Moreover, concessions were made for sunshade and buildings under canopy; conditions that seldom affect urban-

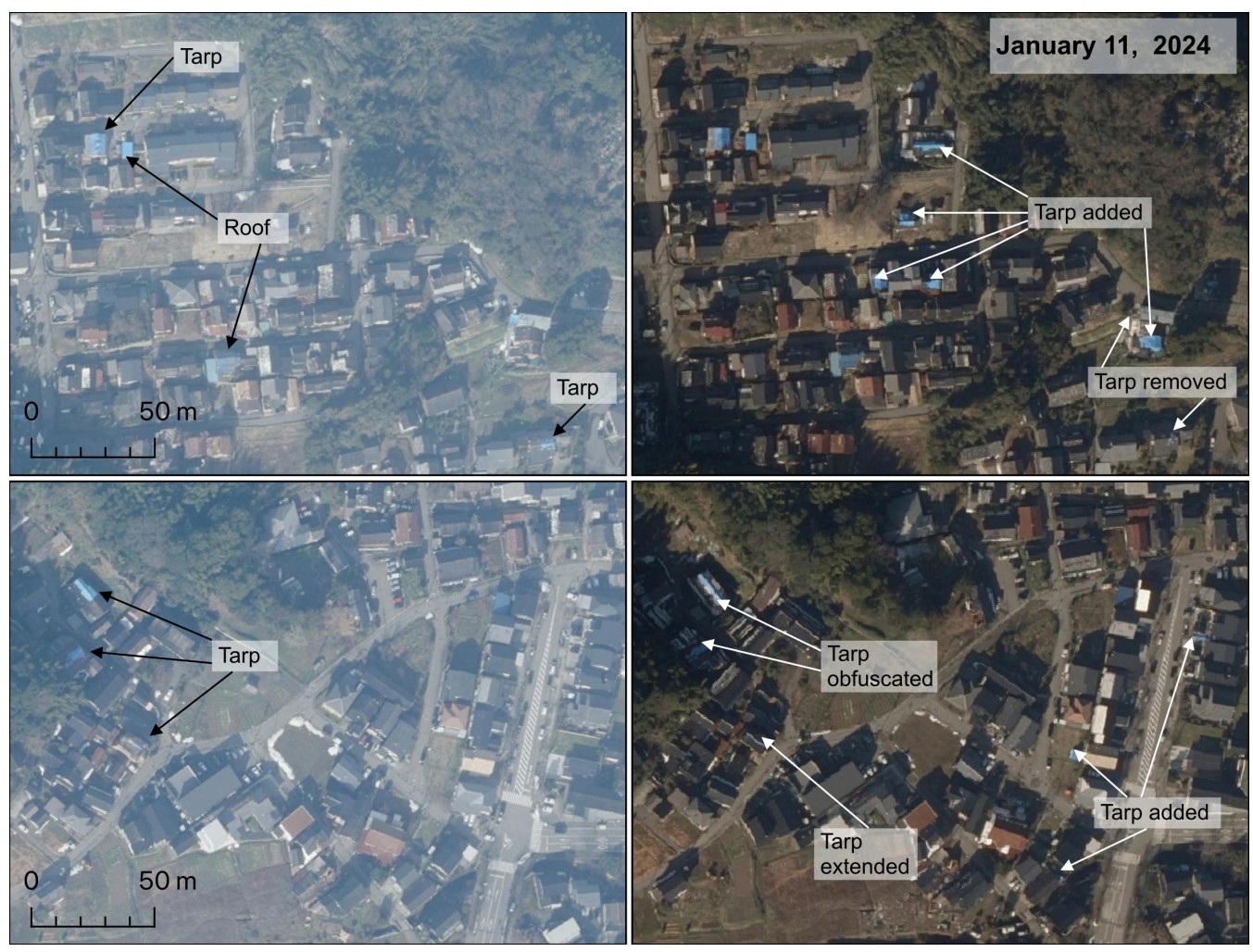

**Figure 6.** Examples of change in blue tarp coverage in Wajima City between aerial imagery capture missions. The figure highlights challenges faced through potential issues in coverage, atmospheric conditions, and source mismatch. Aerial imagery courtesy of GSI (2024).

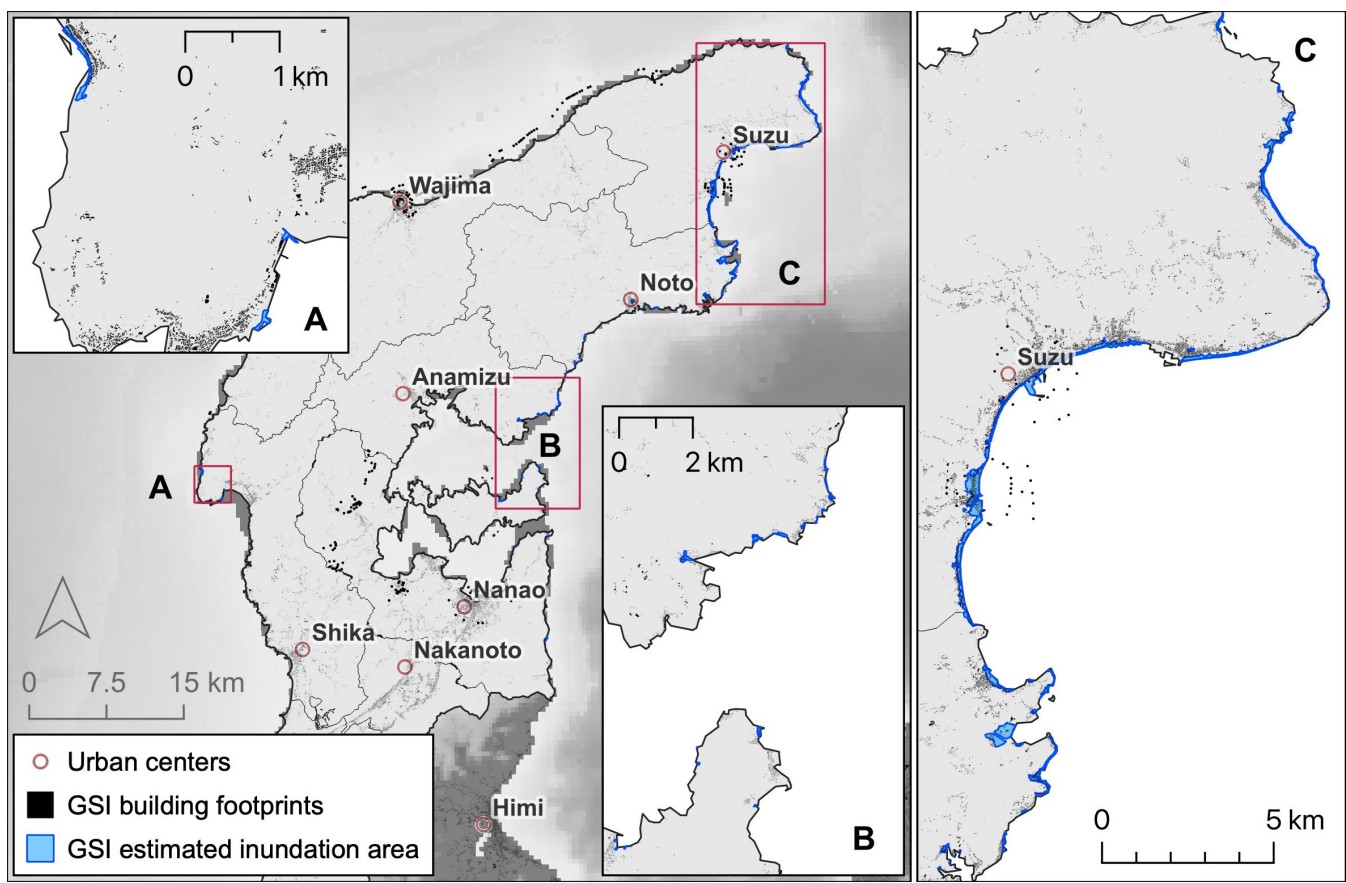

**Figure 7.** Estimated inundation area provided by GSI; backdrop: GEBCO Bathymetric Compilation Group 2023 (2023).

ized coastal settlements where tree cover is diminished. The final damage inventory for the whole domain resolves to 140,208

buildings of which 25,685 were digitized manually. The large proportional disparity in digitized buildings between the tsunami affected areas (4.1%) and the entire domain (18.3%) is largely due to vast portions of building footprints in the countryside being mismatched (Figure 2).

## 2.5    Crowd sourced feedback

An initial version of the database was made public for viewing on **February 11, 2024** https://experience.arcgis.com/ex

perience/70aae9964dc54e4190b6b360dcbb3759/. The working group encouraged specialist opinions to validate potential errors in the data. Corrections are submitted directly through the website and must include photos for the review process to be formalized. Building damage in Nanao City was particularly challenging due to the a combination of poor exposure, vertical image desaturation, and densely packed houses. In this case we relied on news public reporting (xTECH, 2024; Minami, 2024) that included images and location descriptions to identify damaged buildings.

A second set of review information was made available by limited-scope surveys (conducted by research teams) who provided photo evidence to assist the technical validation process. This data informed a quantitative statistical analysis of error margins for the assessment.

## 3  Data Description

In this section we describe the structure of the dataset, technical notes, attributes, and secondary sources. The database is stored as a GeoPackage (Yutzler, 2024) `Noto_Peninsula_Damage_X_Y.gpkg` (where X and Y are version values). A single layer (vX.Y) with table entries for contents (features) and geometries (`MultiPolygon`) is used to store the building footprints. Details regarding each feature are given in Table 5. A total of **140,208** entries (features + geometries) are included in the dataset. The basis of the analysis was conducted on top of GSI (2024)'s "basic map information" publicly available at https://fgd.gsi.go.jp/download/. The raw data is organized in tiles comprising the standard national mesh defined in JIS X 0410:2002 (Japanese Industrial Standard Committee, 2021). Each tile archive (Provided as: `FG-GML-nnnnnn-ALL-YYYYMMDD.zip`, where `nnnnnn` is the mesh tile number and `YYYYMMDD` is the date of the last update). In our assessment we only consider the `FG-GML-nnnnnn-BldA-YYYYMMDD-0001.xml` files which contain the building polygons. We retain only the `geometry` and the `s_fid` attribute from the original dataset. Tiles relevant to the present assessment are given in Figure 8, however we aggregate them into a single table in preprocessing. The dataset uses coordinate reference system (CRS) `EPSG:4326` (WGS 84).

## 4  Technical Validation

The database was split into working subsections to annotate using single-source or multi-source remote-sensing imagery (Figure 5) – we refer to this process as human annotation (Xia et al., 2023). Once completely classified, each subsection was reviewed by a different team member and integrated into the live database. Our Iterative validation was twofold: Through our open web API, we collected voluntary requests for correction, each submission requiring photographic evidence (Figure 4). Each building for which a correction was submitted was given a new validated damage class (Table 5) with the new classification provided that the submitted evidence conformed to our criteria (Table 2).

Data provided by two independent on-site photographic surveys, respectively Tokoha University and Tohoku University, was used to validate portions of the database similar to how crowd-sourced data was handled. The surveys provide coherent coverage of 4 major settlements: Wajima City, Suzu City, Anamizu, and Monzenmachi (Wajima City); as well as scattered inland rural settlements (Figure 9). Since the data is unbiased with respect to the database (i.e., all damaged buildings were documented along the survey path irrespective of the damage assessment class), the coverage was used to statistically impute the accuracy of human annotation: Each photo was taken from ground level and geo-tagged, forming a dense set of nodes. An approximate path was generated using a range-limited nearest neighbor algorithm. Finally, the intersection between the building database and a 40m buffer (reasonable field of view, assumed from photo inspection) around the paths was taken as the surveyed extent. Our initial labels (human annotation) are taken as the estimates $\hat{y}$ and measured against the surveyed

| Attribute | Type \| Length | Valid entries | Description |
|---|---|---|---|
| `fid` | `Int64` | [1 — 140 208] | Unique identifier for the building (original). |
| `s_fid` | `String\|80` | GSI serialization standard or 'manual' | Serial feature identifier from the original xml file GSI (2024); `manual` when manually added. |
| `damage` | `Int8` | [0, 1, 9, 99] | Damage class attributed as part of this assessment, as per Table 2. |
| `damage_val` | `Int8` | [0, 1, 9, 99] | Damage class after technical validation (Section 4) attributed as part of this assessment. |
| `source` | `String\|30` | array or NULL | Oblique image source number from KKC inventory(KKC, 2024) (where available); see Section 6 for access to the KKC repository. |
| `municipality` | `String\|20` | *Prefecture-City-Town* (Japanese) | Municipality name from e-Stat[a] (MIAC, 2024). |
| `conf` | `String\|10` | [single, multi] | Confidence level of the assessment as per Figure 5 based on oblique coverage. |
| `GSI_fire` | `Bool` | [0,1] | Whether building intersects GSI (2024) fire-impacted polygon |
| `GSI_slope_failure` | `Bool` | [0,1] | Whether building intersects GSI (2024) slope failure polygons |
| `GSI_tsunami` | `Bool` | [0,1] | Whether building intersects GSI (2024) tsunami inundation polygons |
| `USGS_MMI` | `float` | Real | Modified Mercalli Index inherited from the USGS (2024) layer |
| `geometry` | `MultiPolygon` | `MultiPolygon[Polygon (…)…]` | Vector geometry of the building footprint (GSI, 2024). |

[a] Available at: https://www.e-stat.go.jp/gis.
Sitemap (JP): トップページ / 統計地理情報システム / 境界データダウンロード
Query tags (JP): 小地域, 国勢調査, 2020年, 小地域 (基本単位区) (JGD2011), 世界測地系緯度経度・Shapefile,石川県

**Table 5.** Details regarding table attributes contained in the GeoPackage dataset.

(corrected) ground truth $y_{GT}$, we report standard classification metrics in Table 6. The harmonic $F_1$-score between survived and destroyed classes is 0.939, suggesting high confidence in the assessment. A spatial representation of the survey coverage is

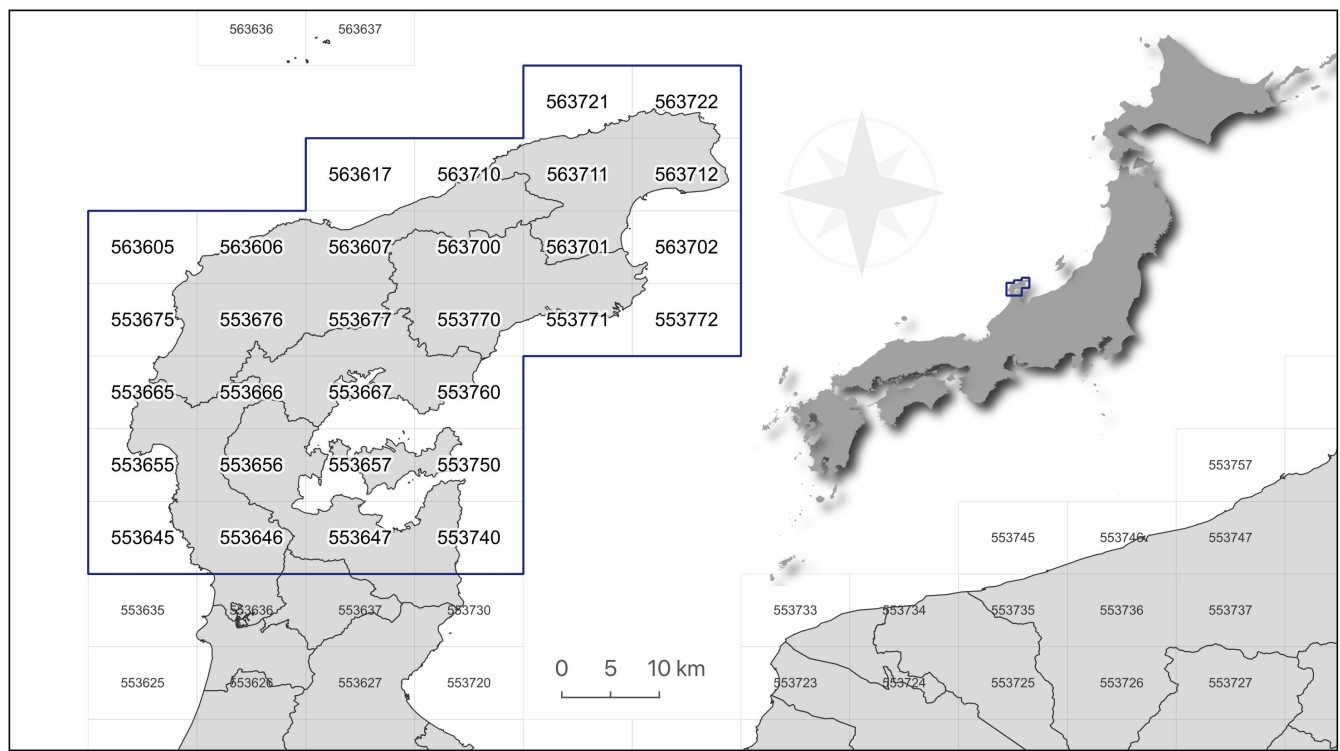

☐ Subset of tiles considered in the assessment

**Figure 8.** GSI mesh tiles considered for the assessment (GSI, 2024), available at: https://fgd.gsi.go.jp/download/.

| Class | Precision | Recall | $F_1$-score | Samples |
|---|---|---|---|---|
| **Survived** | 0.95 | 0.99 | 0.97 | 1666 |
| **Destroyed** | 0.99 | 0.84 | 0.91 | 559 |

**Table 6.** Classification statistics for independently surveyed areas, showing the approximate accuracy of the visual assessment against comprehensive ground documentation.

given in Figure 9, notably a large portion of the surveyed areas are outside of multi-source coverage, suggesting that despite the limitations described above, the proposed visual assessment framework is robust. We hope that this exercise in crowd-sourced

and survey validation will permit further statistical investigations into the features and limitations of manual image-based rapid building damage visual assessments.

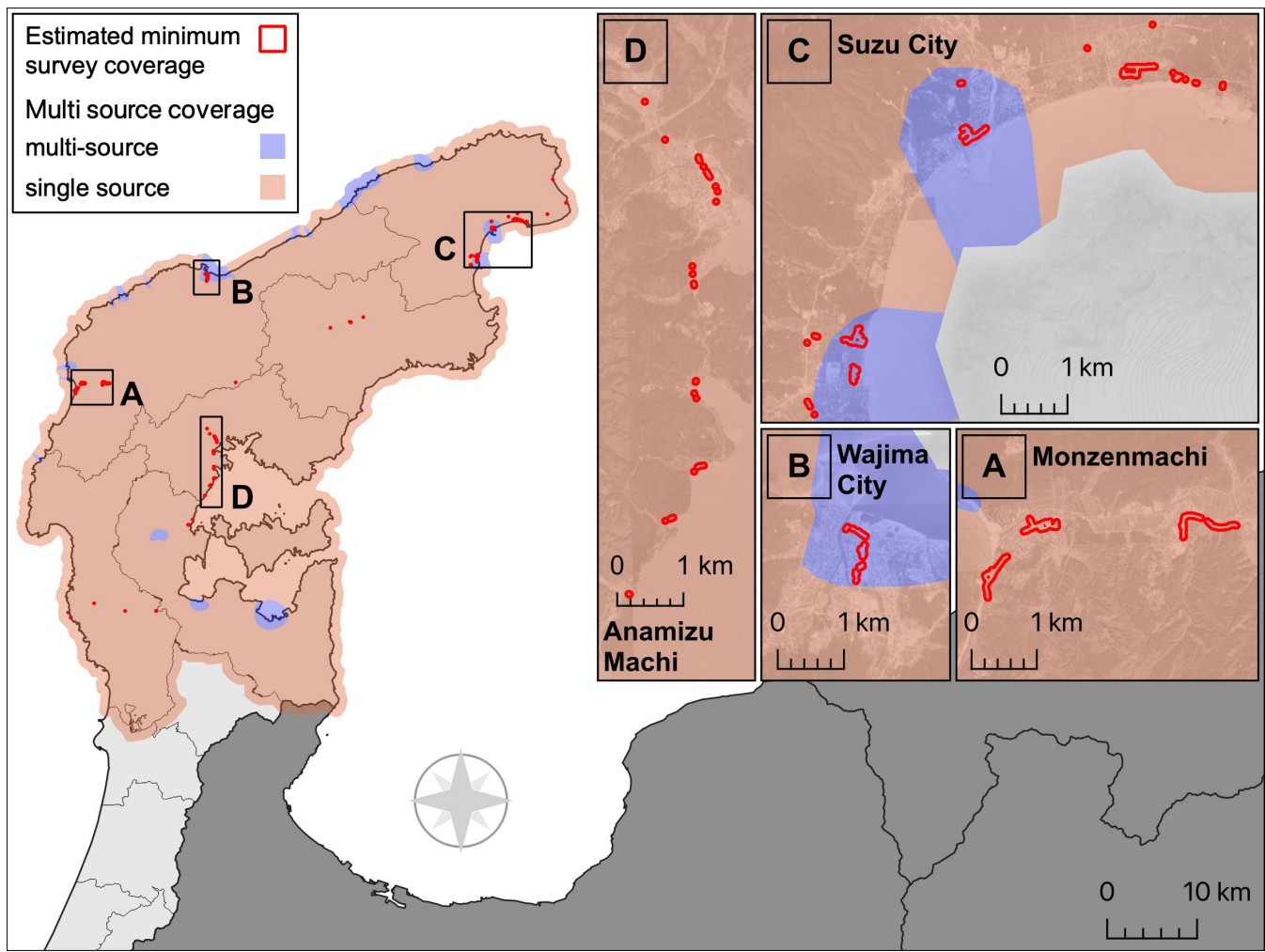

**Figure 9.** Dataset validation areas are estimated from imagery provided by an independent survey team. A path was fit through the location metadata of each photo. We assume a 40 m range buffer around the path as a reasonable visible area for the survey team, judging by the photographic evidence provided. Inset basemap courtesy of © Google 2024.

## 5 Discussion

The unique nature of the disaster is reflected in its varied impacts on buildings, such as: ground shaking, subsidence, uplifting, tsunami surge, soil liquefaction, landslide, fire, among others (Figure 3). The dataset provides a comprehensive visual assessment of building damage across the Noto Peninsula, including all the aforementioned impacts. With this contribution, we aim to provide a reference for future studies and a benchmark for automated methods.

To guarantee a high degree of consistency across all working members, our classes needed to be as clear-cut as they were manageable. A potentially useful third class would necessarily split the "Survived" class, analogous to the scale provided by Copernicus Emergency Management Service (CEMS) (CEMS, 2025). Roof damage and horizontally displaced rubble are generally the only visible signs of a damage spectrum between ideal "no-damage" and "destroyed" classes. Any potential third class would be predicated on the presence of these defining characteristics.

Regrettably, timing and weather conditions severely limited the return period of sufficiently clear or redundant vertical imagery (Table 1). Major seasonal pressure systems accompanied modest seasonal snowfall across the peninsula, over the first 3 weeks of the year. The cloud cover and snow buildup effectively impeded the classification of 9,456 buildings (Criteria details are given in Table 2). Notable among the mosaics described in Table 1 is the `Nanao_2024-01-05` mosaic which is particularly faded and unsaturated.

The winter season poses particular challenges to image classification – much of the natural environment tends towards darker, less saturated colors due to a combination of factors: Houses in the Noto Peninsula generally feature traditional black roof tiles. Overcast weather can decrease color saturation by reducing available light (hence reflection). Relative darkening can reduce the contrast between dark roofs and the background environment, in both cities (concrete & asphalt colors) and countryside (deciduous vegetation tends towards browns and greys). Many of these challenges can hamper the visibility of roof cracks, missing tiles, exposed roof beams, and scattered rubble that may used to distinguish between classification grades.

The Copernicus guidelines for CEMS make a case to diverge from EMS-98 (Grünthal et al., 1998) citing that "such methodologies are fundamentally designed for ground-based field assessments, and thus are not intentionally tailored to be used with remotely sensed images". Moreover, EMS-98 only considers masonry and reinforced concrete buildings, which are inappropriate for a context such as Japan where wooden buildings are overwhelmingly prevalent. Okada et al. (Okada and Takai, 1999) provide a damage index (and equivalent grades) better suited to the context of the Noto Peninsula, however, this index is also conceived for ground-based assessments and relies on accurately assessing the condition of load-bearing walls and pillars. In principle, the CEMS index provides the most appropriate framework for our use case, however CEMS is not designed with a consideration for multi-source, multi-modal data. Obliques can allow for vastly more granular classification contingent on the viewing angle and distance from the target building. The following items contributed to our final decision to use a binary classification:

– Only $\sim 16\%$ of buildings lie within the viewing angle of oblique imagery.

- While oblique imagery provides significant redundancy, not all buildings visible in the frame are clear, draw distance and image resolution are significantly more variable than in vertical imagery where the distance to the target is more consistent.

- The failure modes varies depending on the hazard (e.g., tsunami, landslide, fire, etc., Figure 3) hence some sort of equivalence is needed to compare the different failure modes relative to the same scale.

- Oblique cover is split between failure modes: for example, in Suzu City, earthquake and tsunami damage is present; while in Wajima City, earthquake and fire damage is present; finally along the north coast between wajima and Suzu City, majority of the damage is landslide and slope failure.

Ultimately, we valued consistency and comparability over the potential for a conditional, more granular classification. For the purposes of this project a binary classification was deemed preferable – a breakdown of how our assessment relates to popular reference scales is given in Table 3. We fully endorse and encourage the use of this dataset by the research community and beyond, as the starting point for more granular and detailed assessments of the damage now that significantly more information is available.

## 5.1 On multi-hazard failure modes

The dataset can inform studies that aim to understand the different multi-hazard failure modes given the different impacts listed above — Valentijn et al. (2020) explore multi-hazard damage detection models, but focus on aggregating each hazard discretely by type. However, as Figure 10 illustrates, multi-hazard failures not only occur within the same domain, but can present in contiguous sections of the same town. In the figure, we show how earthquake damage is often compounded by fire, landslide, or tsunami damage; in cases of more populated areas, multiple hazards are present at once, as can be seen in Wajima City where fire, landslide, and earthquake damage are all present.

## 5.2 Machine learning applications

In the field of disaster geo-informatics, our dataset can serve as training data for machine learning tasks. In its current form, the dataset can be used to test pre-trained models such as those proposed by Miura et al. (2020); Deng and Wang (2022); Wiguna et al. (2024a). In this context our dataset offers a new, valuable out-of-domain test set (Wiguna et al., 2024a). A speculative framework, specifically focused on the multi-hazard nature of the Noto Earthquake disaster discussed above, is illustrated in Figure 11. Combined with population data, our database can enable more granular quantitative research into injury and mortality.

## 5.3 Statistical approaches and baseline model

To stoke the research community's engagement, we provide a statistical baseline of the damage across the non-inundated portion of the Noto Peninsula dataset (Figure 12). We propose an aggregated seismic empirical fragility function relative to

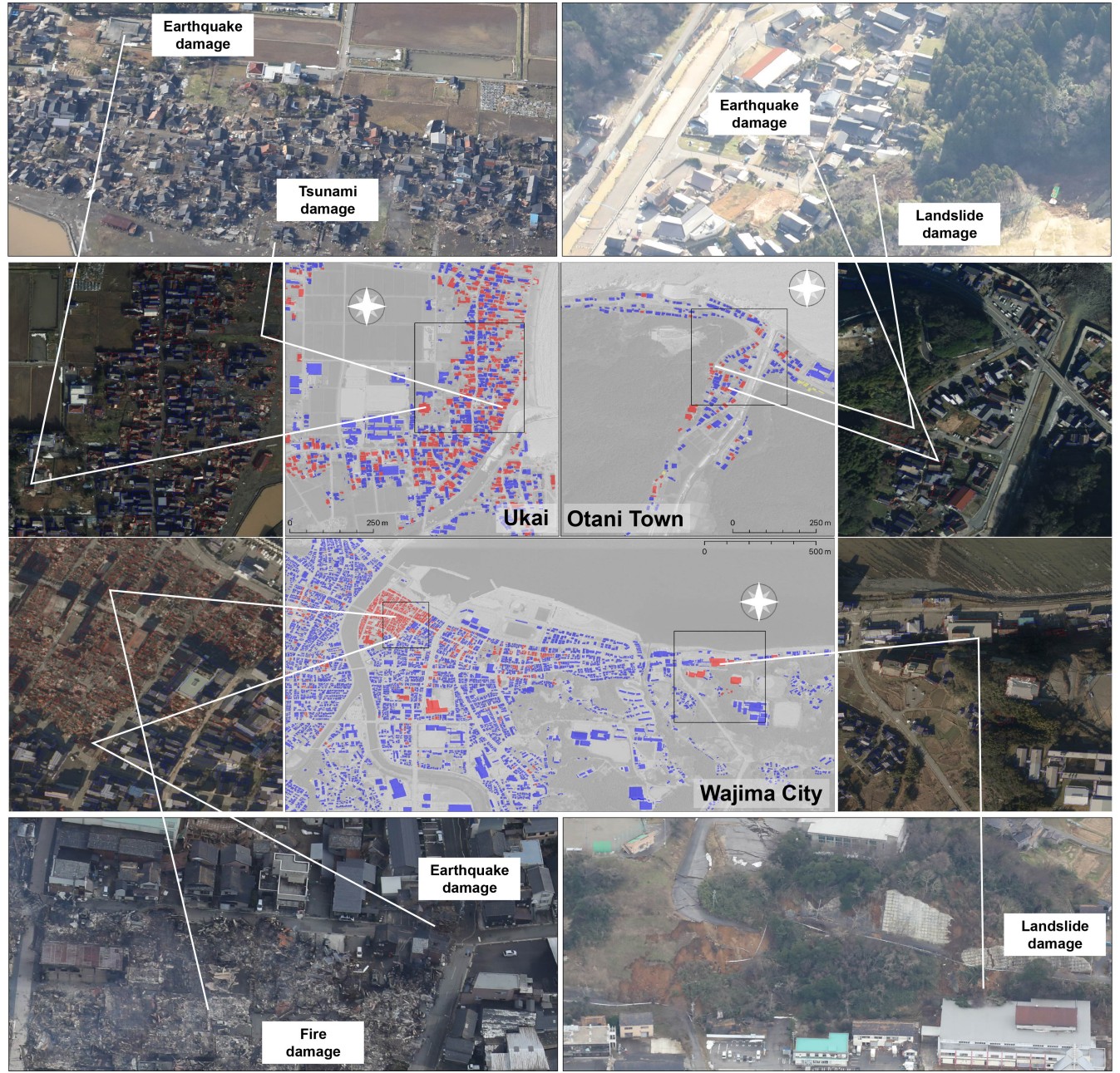

**Figure 10.** Different impacts across contiguous areas of the Noto Peninsula illustrate how multiple hazards may manifest across a single event with extreme proximity. Basemap courtesy of © Google 2024, Obliques by © KKC (2024).

the Peak Ground Velocity (PGV) registered during the event. Importantly, this fragility function is built on the subset of data that was not affected by aggravating hazards (inundation, fire, or landslide) illustrated in Figure 13. Hence we assume that the

## Potential future applications for multi-hazard understanding

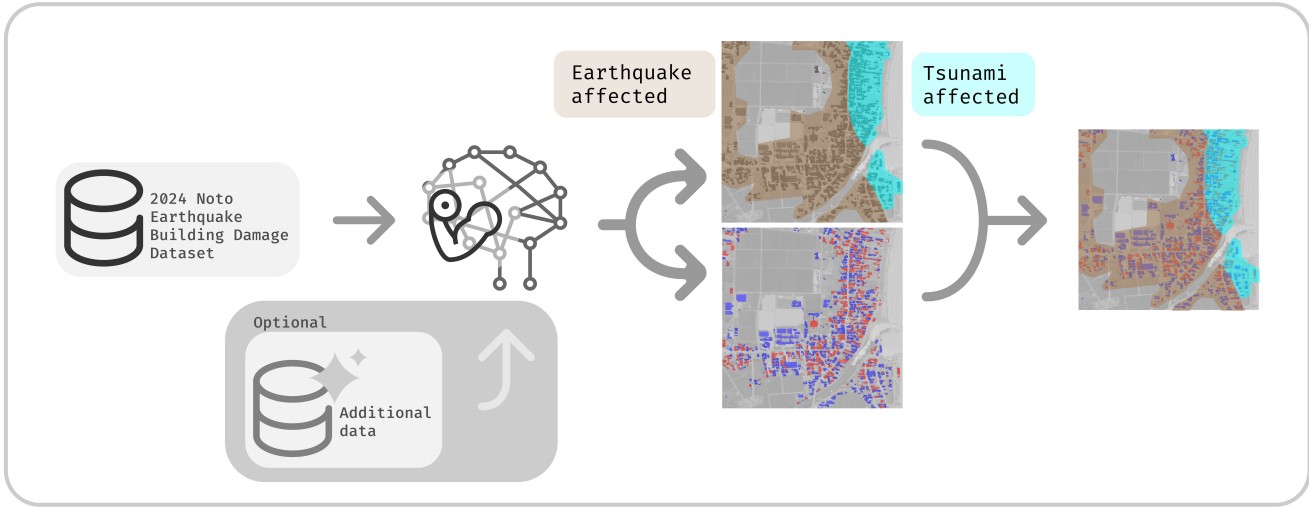

**Figure 11.** A speculative framework that might be used to investigate multi-hazard failure modes (for illustrative purposes). Basemaps courtesy of © Google 2024.

damage is solely due to seismic shaking. We fit the aggregated data using a lognormal distribution (Equation 1) and estimate the parameters using ordinary least squares ($\mu = 6.436, \sigma = 0.9869$).

$$F_X(x) = \Phi\left(\frac{\ln x - \mu}{\sigma}\right) \tag{1}$$

As a frame of reference, we report two fragility functions proposed by Torisawa et al. (2022) for new wooden buildings affected by the Kumamoto Earthquake in 2016. Our baseline fragility function suggests that buildings in the Noto Peninsula were similarly vulnerable to wood buildings built between 2001 and 2016 and destroyed in the Kumamoto Earthquake.

## 6 Data Availability

The database is provided as a standard GeoPackage (Yutzler, 2024) containing a single vector layer accessible through any software implementing the Geospatial Data Abstraction Library (GDAL/OGR) such as QGIS or ArcGIS. Each entry is represented by a building footprint with 7 attributes summarized in Table 5.

1. The database is available in its most updated version at our public repository at (Vescovo et al., 2025) [1],

2. Epicenter and intensity contours are available at the USGS event page [2],

---

[1]https://doi.org/10.5281/zenodo.11055711
[2]https://earthquake.usgs.gov/earthquakes/eventpage/us6000m0xl

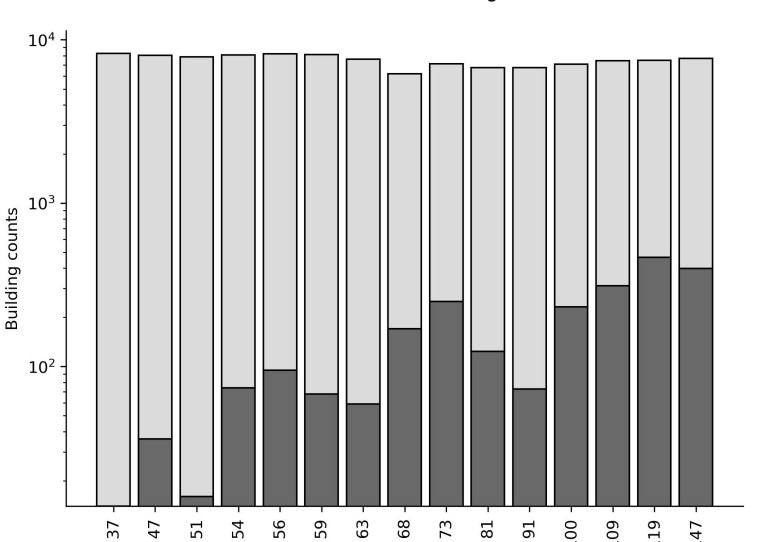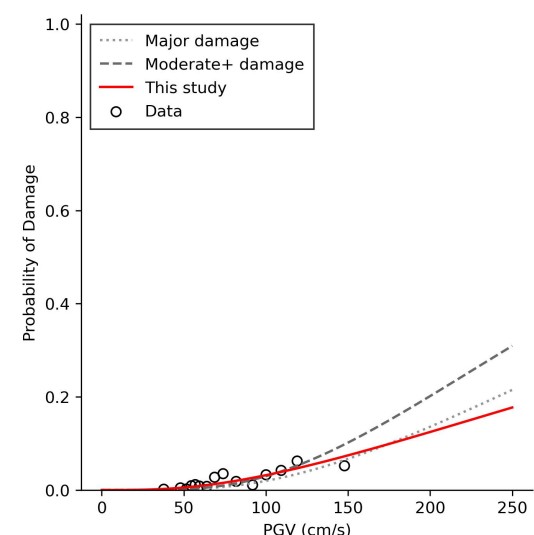

**Figure 12.** Left: Histogram of aggregated building damage. Right: Empirical fragility function (red solid line) for earthquake-affected buildings relative to PGV. We provide two wood buildings fragility functions for "Major" and "Moderate+" damage classes (respectively dotted line and dashed line) proposed by Torisawa et al. (2022) for buildings built between 2001 and 2016 and affected by the 2016 Kumamoto Earthquake.

3. Earthquake swarm data is available through the Japan meteorological Agency (JMA)'s website [3],

4. Post event raster orthophotography, inundation, fire, and slope failure vector extents are available through the GSI's dedicated Noto Peninsula Earthquake page (GSI, 2024) [4],

5. Oblique imagery is provided by KKC (KKC, 2024). For disaster events, KKC may make their products available for free through the BOIS portal [5], subject to terms and conditions [6]. With permission from KKC, only oblique images freely available through BOIS were employed in the present study.

## 7 Conclusions

We present a comprehensive building damage database for the Noto Peninsula Earthquake of 2024, developed through a multi-source, multi-modal visual assessment of building damage. The particular circumstances of this event, timeliness of data avail-

---

[3] https://www.data.jma.go.jp/eqdb/data/shindo/

[4] https://www.gsi.go.jp/BOUSAI/20240101_noto_earthquake.html, Japanese.

[5] https://bois-free.bousai.genavis.jp/diarsweb, Japanese.

[6] https://www.kkc.co.jp/contact/image/, Japanese.

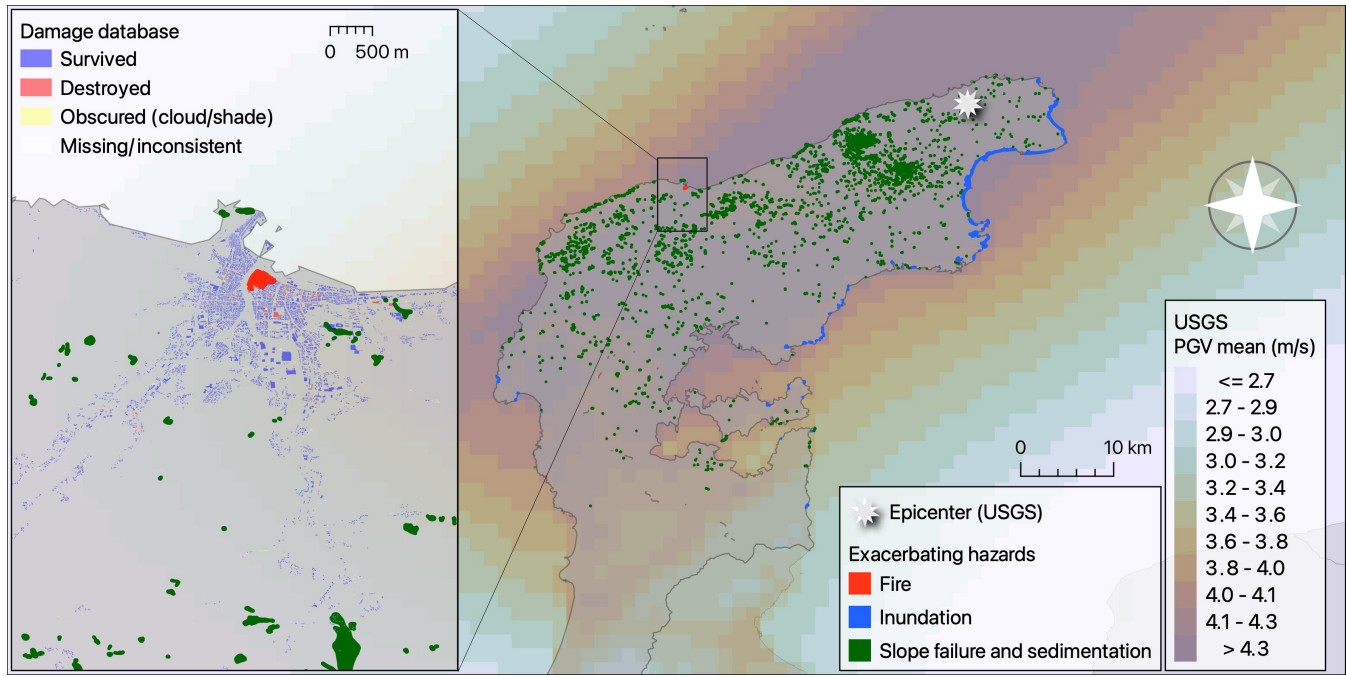

**Figure 13.** Composite of exacerbating hazards (in addition to seismic impact). Inundation, fire area, and slope failure/sedimentation extents are provided by the GSI (2024). Peak ground acceleration estimates provided by KKE (2024) in collaboration with NIED (2019).

ability, degree of coverage, and access to in-situ survey information, presented a singular opportunity to develop and validate this new dataset through a unique framework. By providing this dataset offers the opportunity to study impacts of multi-hazard disasters on building damage. Figure 10 illustrates how different hazards manifested across contiguous areas of the Noto Peninsula. Understanding the different impacts may provide valuable insights to disaster response and recovery planning. Future studies may leverage our dataset to develop novel multi-hazard models that can predict building damage across different impacts (A speculative framework is shown in Figure 11). With this contribution we hope to enrich the global corpus of disaster building damage datasets. We provide the hand curated building inventory as a GeoPackage through the public repository at https://doi.org/10.5281/zenodo.11055711 (Vescovo et al., 2025). Each building was classified into 4 classes: Survived, Destroyed, Obstructed view through human inspection, and Missing or inconsistent. Limited scope validation was conducted through crowd-sourced community feedback through our online portal and independent survey data conducted by experts in the field. In its immediate form, the dataset may be used to:

– train site specific statistical and machine learning models for building damage assessment.

– test domain adaptation frameworks for building damage assessment by testing pre-trained models on our new out-of-domain dataset as illustrated by Wiguna et al. (2024a).

– fine-tune pre-trained models on our dataset to improve performance across datasets as shown in Wiguna et al. (2024b).

– develop novel multi-hazard models that can predict building damage across different impacts.

In combination with additional data source, such as population data, and post disaster information, our dataset can inform further investigation into disaster logistics, evacuation, injury, and mortality. We hope that this dataset will serve as a reference for future studies on building damage assessment, disaster response, and recovery planning.

**Author contributions**

R.V. Writing — original draft preparation, Validation, Visualization.

R.V. and S.W. Methodology, Software, Formal analysis, Data curation, Investigation.

CY.H., J.M., X.D., S.I., K.W., and Y.E. Data curation.

B.A., E.M., and A.M.: Conceptualization, Investigation, Supervision, Field Survey, Writing — review & editing.

S.K.: Conceptualization, Supervision, Field Survey, Writing — review & editing, Funding acquisition.

*Competing interests*.  The authors declare that they have no conflict of interest.

*Acknowledgements*.  This study was partly supported by JSPS KAKENHI (Grants-in-Aid for Scientific Research, 21H05001, 22K21372, and 22H01741) and the Cross-Ministerial Strategic Innovation Promotion Program (Grant Number JPJ012289). The use of building footprint data is approved by Geospatial Information Authority of Japan (GSI) (R5JHs641).

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
