# Peer review of "The 2024 Noto Peninsula earthquake building damage dataset: Multi-source visual assessment"

_Earth System Science Data, 2024_

## Author Comment (AC2)

Response to Referee 2 on ESSD-2024-363

First and foremost, we would like to thank Referee 2 for their encouraging comments and thorough feedback. Below I have included a point by point response. I would invite the referee to also review the edited manuscript to verify that their queries were answered in a satisfactory manner.

**Major Suggestions**

1. Contextualization within existing damage datasets

While the authors provide a useful comparison of classification schemes in Table 5, I encourage them to elaborate further on how their dataset compares with other similar open-access post-disaster damage datasets (e.g., in the Japanese context, the dataset compiled by the Ministry of Land, Infrastructure, Transport and Tourism (MLIT) for the 2011 Great East Japan Earthquake). Key aspects to highlight might include the attributes provided, accessibility for users, the level of detail in hazard-specific classification, and the potential for reuse in modeling or planning. A broader contextualization would help readers better appreciate the potential of this new dataset and identify synergies with existing efforts.

The 2011 dataset achieved such levels of detail following several structured surveys of the affected areas across the 6 prefectures impacted. While in principle we agree that such information would be invaluable it is the prerogative of MLIT to conduct and release this information, which, at time of writing has not been made available to us. Because the dataset is based originally on GSI polygons, we are at best limited to the information embedded in the original footprint, with regard to the requested attributes.

In terms of contextualization, I have made several changes to the introduction and added a more concrete specification of why our dataset is relevant to the field and what contribution we hope to provide.

2. Multi-hazard characterization and enrichment of attributes

Given the multi-hazard nature of the Noto Peninsula event (earthquake, tsunami, landslides, fires), it would be highly valuable to incorporate hazard-specific indicators directly within the dataset. Although links to external hazard data are provided (p. 19), embedding this information at the building level (e.g., presence of tsunami inundation, fire impact, or landslide proximity) would significantly enhance usability, especially for non-Japanese users. For instance, the mentioned 2011 MLIT dataset includes information on water depth indicators and indication for the concurrent presence of other hazards. At minimum, a discussion on the feasibility of such integration and its relevance for downstream applications would enrich the paper.

Referee 1 also expressed interest in this item. Indeed we recognize that this a critical aspect of any furhter multi-hazard analysis would greatly benefit from this information. The requested attributes have been added to the database and an updated copy (and metadata) has been uploaded to zenodo (v2.5). Consequently, table 5 has been updated to reflect the additional attributes:

| Attribute | Dtype | Description |
|---|---|---|
| GSI_fire | Bool | Whether building intersects GSI (2024) fire-impacted polygon |
| GSI_slope_failure | Bool | Whether building intersects GSI (2024) slope failure polygons |
| GSI_tsunami | Bool | Whether building intersects GSI (2024) tsunami inundation polygons |
| USGS_MMI | Float | Modified Mercalli Index inherited from the USGS (2024) layer |

While the above was added for each building, without a forensic survey, it is difficult to make a value judgement on the single most likely hazard that contributed to the damage, with any degree of finality. We hope this addition achives a good level of compromise.

3. Discussion improvements

The discussion section could benefit from a more detailed reflection on the limitations and strengths of the dataset with respect to different user communities (e.g., researchers, emergency managers, machine learning practitioners). It may also be worth clarifying how the dataset could evolve in the future, for instance by incorporating additional damage explicative features, higher-resolution classifications, etc.

Well noted, the discussion section has been significantly altered following feedback from both reviewers, including an acknowledgement for further developments that build upon the dataset.

**Minor revisions**

Some minor reordering of figures and tables based on their first appearance in the text would improve readability.

Well noted. Figures have been reordered to appear in order

In the current PDF version of the manuscript, mathematical symbols and equations do not render correctly and are therefore not visible to the reader

There seems to be an issue with the post-processed version of the manuscript on the Copernicus side. On my side of the MS records, I can confirm that symbols are rendered in the version of the manuscript that was submitted, the issue appears to be only on the Copernicus preprint version. This does not seem to be a latex issue since at this stage we were asked for a precompiled version rather than the source files.

L27-29: This sentence seems grammatically incomplete, as it lacks a main verb.

Well noted. The sentence was indeed incomplete and has been amended.

L41: A full stop should be inserted after 'building damage'.

Well noted.

To improve the logical flow of the paragraph, it may be helpful to relocate L96-98 ('We provide [...] middle classes') immediately after L91 ('[...] Table 1').

Well noted, the text was rearranged as suggested.

L94-95: this sentence is not clear, please consider rewriting.

Well noted. The sentence itself and the following sentence were restructured to be clearer.

Figure 5: Increasing the line thickness of the building polygons would likely improve visual clarity.

Well noted. Line thickness was increased.

L153: A full stop should be inserted after 'original dataset.

Well noted.

L168: missing 'of' between 'environment' and 'both'

Well noted.

I would like to express again my appreciation for the time and attention that the referee dedicated to the manuscript. I hope I was able to answer in a satisfacotry manner through this direct response to comments and in text.

---

## Author Comment (AC3)

Response to Referee 1 on ESSD-2024-363

First and foremost, we would like to thank Referee 1 for their kind comments and thorough feedback. I would like to take this opportunity to respond point by point. I would also infite the referee to review the final manuscript and verify whether their queries were duly addressed.

I missed a short explanation of why the most probable hazard type(s) are not included, because in this case study there are so many different hazard types at play which makes it very interesting. It is likely that this was just too difficult, but please mention that, even more so because in Figure 5 you make a distinction between earthquake/tsunami/landslide/fire. It could even be an idea to put the multi-hazard nature in the title, because it may be interesting for others too.

Indeed, this would be a critical aspect of any work investigating the multi hazard nature of the event. Both yourself and Referee 2 showed interest in this specific point. To that effect, we have uploaded a new version of the database (v2.5) to the Zenodo repository. This new version has 4 new attributes:

| Attribute | Dtype | Description |
|---|---|---|
| GSI_fire | Bool | Whether building intersects GSI (2024) fire-impacted polygon |
| GSI_slope_failure | Bool | Whether building intersects GSI (2024) slope failure polygons |
| GSI_tsunami | Bool | Whether building intersects GSI (2024) tsunami inundation polygons |
| USGS_MMI | Float | Modified Mercalli Index inherited from the USGS (2024) layer |

While we are not able to make a value judgement of a single most likely hazard, we include the seismic exposure magnitude in Mw for each building, and boolean values for each of the other hazards highlighted in figure 5. We hope this addition achives a good level of compromise.

**Specific comments**

**Introduction**: The introduction starts with an extended description of the Noto Peninsula earthquake and the cascading hazards following the earthquake. A reference to assessment of (specifically) building damage is only found in line 41. As this article is mainly a data paper, it is my suggestion to keep the description of the hazard very brief. More interesting it is to talk about:

Broader context of building damage datasets (similar to the text following L41).

What other previous building damage datasets are already there. Nepal 2015 is an interesting case because crowd-sourcing has also been used, but in a different way, e.g. https://ieeexplore.ieee.org/abstract/document/7427206, but also L'Aquila 2009 has been well-documented. You also mention Haiti and Christchurch in the method section that could be moved here.

What is the benefit of specifically your dataset

Well noted, the Introduction has been shortened, focused more on damage, and supplemented with information about similar approaches, including the ones reccommended by the referee. In addition the mentions of Haiti and Christchurch were moved here.

Moreover I added some explicit contributions specific to this dataset.

**Methods**: The methods are well-described in the text. However, it could benefit from a better structure. For example, it is not clear what is the point of the `basis for the assessment` paragraph. It is also confusing that the section `Earthquake damage assessment` only contains a description of the case study area, while the `Tsunami damage assessment` also contains other information (like how mismatches are handled). Some suggestions:

Add a short summary of the steps taken in the general methods paragraph (L68-69)

Have a structure similar to (1) data sourcing; (2) building footprint selection (sub-paragraph for tsunami/earthquake) (3) Classification system (4) Verification / Feedback (crowd-sourced + experts)

In the methods, it was also not clear who exactly did the initial damage classification. I assume it was your team?

Well noted. The `basis for the assesment` was renamed `General Methods`, I also added a itemized summary for the assesment process as reccommended.

I added some clarification regarding who conducted the assessment and moved the description of mismatch handling to this section (removed from the Tsunami section) .

**Technical validation**: The first paragraphs of the validation section (L156-170) would fit better in the discussion.

This is well noted, the whole paragraph has been move to the discussion.

**Tables and figures**: The tables and figures are not ordered by first reference, but are scattered through the paper. Please order by first reference.

Well noted. Figures and tables have been reordered by first mention.

**Technical corrections**

L3 `Vector polygons` is redundant. Please just use `polygons`.

Well noted.

L8 `Disaster dynamics models` unclear what this means

The lack of clarity is well noted, changed to `disaster-specific physical dynamics models` - examples would be finite element seismic models that aim to simulate structural collapse, full 3D tsunami inundation flow models, etc.

L11;L99;L187;L219: Formulas and mathematical symbols are not correctly exported / not readable.

There seems to be an issue with the post-processed version of the manuscript on the Copernicus side. On my side of the MS records, I can confirm that symbols are rendered in the version of the manuscript that was submitted, the issue appears to be only on the Copernicus preprint version. This does not seem to be a latex issue since at this stage we were asked for a precompiled version rather than the source files.

L36 `in this capacity ...` Sentence doesn't work

Well noted, this has been corrected.

L42 `multi-source` is a key concept, but never formally explained

Well noted, I have added clarification in brackets at this first mention (excluding title and abstract).

L56 What do you mean with a `baseline dataset`

This has been changed to just `dataset`

L59 As the KKC dataset is proprietary, do you know the license of the data and if it can be used for this purpose? If so, please state here, in the methods or in the data availability

Regarding disasters, KKC makes certain data available through their free BOIS portal. The information is specifically released to assist with "recovery and reconstruction activities" and to "confirm hazard information". For transparency, I provide the original Japanese as found at their webside and my best translation below:

> **OR**: 撮影箇所が地図上にプロットされ、地図上の位置や地形と合わせて現地の状況が一目で判るため、復旧・復興活動に役立てることができます。さらにハザード情報などのコンテンツも併せて確認することができます。※コンテンツは順次公開予定

> **TR**: "Image capture locations are plotted on a map, making it easy to see the local situation at a glance, including the position on the map and the surrounding terrain, which is useful for recovery and reconstruction activities. Furthermore, you can also check content such as hazard information. Content will be released sequentially."

KKC requires that users make a formal request https://www.kkc.co.jp/contact/image in case they wish to employ the imagery provided through BOIS. Additionally KKC reserves the rights to charge a fee depending on the use-case.

L55-L66 there is no mention of the media photos that are mentioned in the abstract

In the abstract we mention news reporting imagery, these sources are given in lines 96 - 99 [formerly 85-86] (Minami 2024, Nikkei xTECH 2024). As of April 9, the articles are still accessible through the links in the references, however we keep a pdf version of the articles for posteriority.

Figure 2: what does multi-modal mean here?

Multi-modal refers to the modality of the media we employed to perform our classification: in the example given modality 1 is the orthophoto (GSI), and modality 2 is the oblique (KKC). Each modality adds to our analysis, for example it can me hard to distinguish some collapse if it's relatively clean and vertical, the oblique helps identify these edge cases, and allows us to appraise lateral damage. Multi-modal is congruent with multi-source in this this case, since we worked with only two sources each providing one mode, but it is not necessarily mutually exclusive.

L88 Tsunami-affected areas are introduced in the next section, why are they mentioned here

Well noted. The project began as a tsunami-only investigation and expanded beyond that later, in the `Basis for the assessment` section we aimed to provide a rationale regarding the inception of the project. I have changed L88 to better reflect this.

L102 `Although this … vertical images`: this sentence is unclear, what do you mean?

This sentence has been changed to be more clear. The crux of the argument (exemplified in Figure 7) is that satellite images that have some (spatial overlap, but were taken on different days show different blue-tarp coverage: for example some tarps are apparently removed, many are added, some become indistinguishable. We considered adding a separate class for tarp-covered strictures (beyond 0 and 1) as indicated by Miura et al. (2020) but we were not able to guarantee consistency.

Figure 8: Hard to read. It's difficult to see the blue/red differences (and what do they mean? Are mismatches in red?)

Well noted. In the original right pane the colors correspond to the classes (0, 1, 9, 99) respectively blue, red, yellow, and white. While in the left pane we show the original polygons provided by GSI. The intention was to show the difference.

In hindsight, as pointed out, the colors do not serve the purpose of the figure. I have modified the figure to use a binary color scheme: yellow to indicate the original GSI polygons and Purple to indicate the polygons corresponding to buildings that actually existed in the most recent pre-disaster imagery. Purple polygons consist of either: formerly yellow polygons that actually existed in the pre disaster imagery or new polygons hand-drawn and based on the pre-disaster imagery.

I have also changed the base-map to use a grayscale color scheme and increased the contrast to improve clarity.

L240 the statement here counters the statement made in L189. If your method is robust also in areas without multi-source input, why would it be better than just a single-source approach?

Absolutely. The statement in L240 mischaracterizes the study and has been corrected, we thank the referee for pointing out the oversight.

To clarify: It is not our intention to claim that the present method is in any shape or form superior to other methods.

Rather, we hoped to show that, by having access to multi-source information and survey information, we have the ability to check that our classification appears to be robust across the domain. This gives us a relatively greater degree of confidence in areas that are single source.

Moreover, it must be pointed out that, our results are predicated on the specific data that was available to us for this project, the capabilities of our team, and subsequent survey information that aided our validation.

I would like to express again my appreciation for the time and attention that the referee dedicated to the manuscript. I hope I was able to answer in a satisfacotry manner through this direct response to comments and in text.

---

## Author Response (AR2)

Response to Referee 1 on ESSD-2024-363

We would like to again thank Reviewer 1 for their thoughtful and thorough follow-up. We also appreciate the graciousness of Reviewer 1's evaluation scores, which are very encouraging.

Below we address each point raised:

**Technical corrections:**

**Point 1:** GSI is mentioned in Table 1 and Figure 2, before explanation of the abbreviation in the text on page 5. Please either use the full name, or mention earlier in the text.

Noted. This has been amended.

**Point 2:** In some cases in the text, the reference to earlier figures or tables is rather more confusing than that it helps understand. Reconsider if (1) the figure/table reference really supports the point you make in the text and (2) the point you make in the text needs a figure/table as reference. For example, but not limited to, L186-L191 references to Figure 4 / Table 2. This could for example be changed to ": Through our open web API, we collected voluntary requests for correction, each submission requiring photographic evidence. Each building for which a correction was submitted was given a new validated damage class (Table 5) with the new classification provided that the submitted evidence conformed to *our binary damage classification*."

Very poignant – in hindsight I see what the reviewer is referring to. I have tried to follow the example to remove references that would suggest the reader's attention to jump to a previous figure or table and disrupt the reading process.

A few notable exceptions – I have refrained from removing section references, as I feel that section references are more of an acknowledgement that something is discussed later, or had been discussed above, and less of a suggestion that the reader follow through with reading the section before coming back.

In the data availability section, there is a reference to Table 5. While I acknowledge that this is similarly disruptive, I am of the opinion that this specific reference is less negotiable.

In the conclusion there are references to Figure 10 and Figure 11. I think these are somewhat necessary references to drive the points made in the conclusion.

**Point 3:** Capitalization of lists is inconsistent. L121-L131 for example. L301-L309 are capitalized, L375-379 are not

Noted. This has been amended.

---

## Author Response (AR3)

**Response to the Topic Editor Report (ESSD-2024-363)**

Dear Kirsten,

Thank you for getting back to us so promptly.

I have made the required change. I also took out the Zenodo DOI from the footnotes in the data availability section as Per Mario Ebel's comment:

*Notification to the authors:*
*With the next revision, please transfer the DOI link https://doi.org/10.5281/zenodo.11055711 itself from*
*footnotes to the in-text citation (Vescovo et al., 2025) in the section "Data Availability".*

A few of points:

- In the diff file there seems to be an issue with displaying the changes to the footnote (L:282 in the diff file and L:282 in the updated version of the manuscript)
- I have added a grant to the acknowledgement section at the end.
- I added a slight variation to the manuscript as supplement: **version 5_1**, in which I got rid of **all** footnotes in the "Data Availability" section (rather than just the one pointed out by Mario), hoping to anticipate potential issues with footnotes.
  This version is otherwise equal to the submitted manuscript version (version 5). I provide this as an option but leave the final decision regarding what version to use up to your discretion.